# Validating organoid-derived human intestinal monolayers for personalized therapy in cystic fibrosis

Liron Birimberg-Schwartz[1,2], Wan Ip[2], Claire Bartlett[2], Julie Avolio[2], Annelotte M Vonk[3,13], Tarini Gunawardena[4], Kai Du[4], Mohsen Esmaeili[5], Jeffrey M Beekman[3,13], Johanna Rommens[5,6], Lisa Strug[5,7,8,9], Christine E Bear[4,10,11], Theo J Moraes[2,12], Tanja Gonska[1,2]

**Highly effective drugs modulating the defective protein encoded by the CFTR gene have revolutionized cystic fibrosis (CF) therapy. Preclinical drug-testing on human nasal epithelial (HNE) cell cultures and 3-dimensional human intestinal organoids (3D HIO) are used to address patient-specific variation in drug response and to optimize individual treatment for people with CF. This study is the first to report comparable CFTR functional responses to CFTR modulator treatment among patients with different classes of CFTR gene variants using the three methods of 2D HIO, 3D HIO, and HNE. Furthermore, 2D HIO showed good correlation to clinical outcome markers. A larger measurable CFTR functional range and access to the apical membrane were identified as advantages of 2D HIO over HNE and 3D HIO, respectively. Our study thus expands the utility of 2D intestinal monolayers as a preclinical drug testing tool for CF.**

## Introduction

Cystic fibrosis (CF) is caused by variants in the *CFTR* gene resulting in impaired anion transport across epithelial barriers and consequently to multi-organ disease including the lungs, liver, and gastrointestinal tract (O'Sullivan & Freedman, 2009). CFTR gene variants are grouped into six different classes according to their common molecular defect affecting translation, processing, gating, conduction, and abundance of the CFTR protein (Welsh & Smith, 1993). Over the last decade, drugs targeting specific CFTR molecular defects have dramatically expanded the treatment options for people with cystic fibrosis (pwCF). These drugs promise a pharmacological "cure" for CF disease as they have shown significant improvement of lung disease accompanied by a decrease in the sweat chloride concentration (SwCl), a well-established clinical marker of CFTR function (Ramsey et al, 2011; Wainwright et al, 2015; Donaldson et al, 2018; Keating et al, 2018; Heijerman et al, 2019). Current CFTR modulator drugs target specific gene defects by (a) chaperoning misfolded CFTR to the apical membrane, "corrector" drugs (Van Goor et al, 2011), (b) increasing the open probability of the CFTR channel, "potentiator" drugs (Van Goor et al, 2009), and (c) stabilizing the CFTR mRNA transcript, "amplifier" drugs (Molinski et al, 2017; Dukovski et al, 2020). Combinations of these drugs achieve higher CFTR modulating effects (Mall et al, 2020).

Next to the most common *CFTR* gene defect, c.*1521_1523delCTT (F508del)*, found in 85% of patients, the most of the identified *CFTR* gene variants occur at very low frequencies. This precludes clinical trials aimed at approving modulator therapy for every CF patient. Instead, rare *CFTR* gene variants are subjected to "theratyping," a technique that examines responses to various CFTR modulator drugs in vitro to evaluate individual molecular defects, whereas assaying for the most efficacious drug combinations (Arora et al, 2021; Veit et al, 2021b). This can be done in heterologous expression systems or primary, patient-derived cell cultures. The US Food and Drug Administration (FDA) and the European Medical Authorities (EMA) approve label extension of existing CFTR modulators based on data generated by this technique, in the absence of relevant clinical studies (Durmowicz et al, 2018; Ponzano et al, 2018). Two main preclinical testing platforms have been established, one is based on human nasal epithelial (HNE) cells generated from patient nasal brushes (de Courcey et al, 2012; Guimbellot et al, 2017; Pranke et al, 2017) and the second is 3D human intestinal organoids (3D HIO) derived from rectal biopsies (Dekkers et al, 2016).

[1]Department of Paediatrics, Division of Gastroenterology, Hepatology and Nutrition, University of Toronto, Toronto, Canada [2]Translational Medicine, The Hospital for Sick Children, Toronto, Canada [3]Regenerative Medicine Utrecht, University Medical Center, Utrecht University, Utrecht, The Netherlands [4]Programme in Molecular Medicine, The Hospital for Sick Children, Toronto, Canada [5]Program in Genetics and Genome Biology, The Hospital for Sick Children, Toronto, Canada [6]Department of Molecular Genetics, University of Toronto, Toronto, Canada [7]Biostatistics Division, Dalla Lana School of Public Health, University of Toronto, Toronto, Canada [8]Department of Statistical Sciences and Computer Science, University of Toronto, Toronto, Canada [9]The Centre for Applied Genomics, The Hospital for Sick Children, Toronto, Canada [10]Department of Physiology, University of Toronto, Toronto, Canada [11]Department of Biochemistry, University of Toronto, Toronto, Canada [12]Department of Paediatrics, Division of Respiratory Medicine, The Hospital for Sick Children, Toronto, Canada [13]Department of Pediatric Pulmonology, Wilhelmina Children's Hospital, University Medical Center Utrecht, Utrecht University, Member of ERN-LUNG, Utrecht, The Netherland

Correspondence: Tanja.Gonska@sickkids.ca

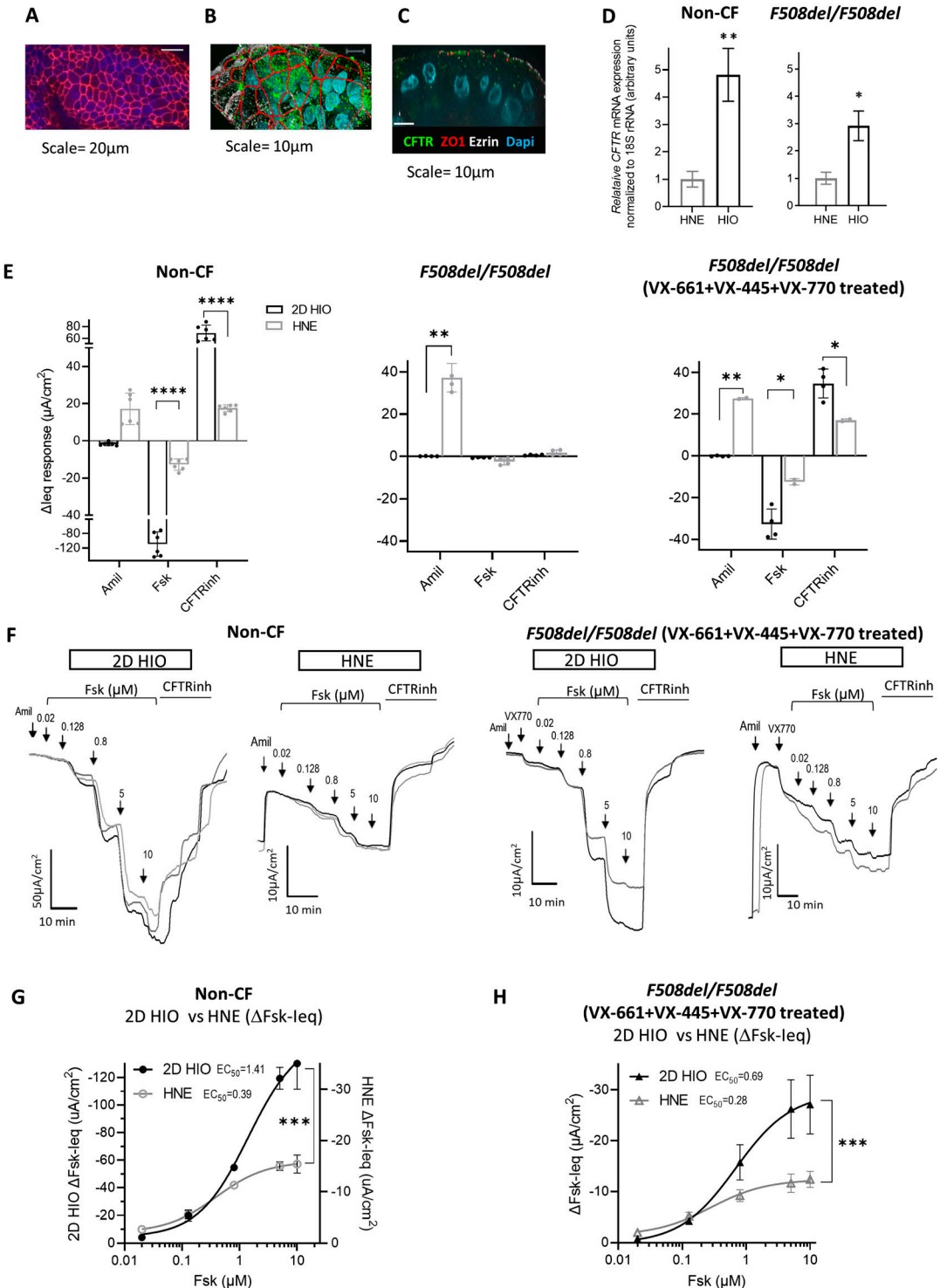

**Figure 1. Epithelial and electrophysiological characteristics of 2D intestinal organoids (2D HIO).**
**(A, B, C)** Immunohistochemical imaging of 2D HIO 4 d post-seeding from a non-CF subject in en face (A, B) and cross section view (C). Immunohistochemical stains demonstrate a tight epithelial layer (ZO1 stained for epithelial tight junction), with polarity (ezrin stain for the apical cell membrane) and apical localized CFTR channels. **(D)** The bar graph shows the comparison of CFTR mRNA expression levels between 2D HIO and HNE from a non-CF subject and an *F508del/F508del* CF patient. *CFTR* mRNA expression was normalized to18S rRNA. The normalized mRNA levels are shown as fold levels after setting the HNE mRNA levels of one sample arbitrarily at 1. Mean ± SD were obtained (two biologic replicates for HNE, triplicates for 2D HIO). Statistical comparison: two-sided unpaired *t* test, *P < 0.05, **P < 0.01. There was no statistical

Methodologies for assessment of CFTR function and drug responses differ between the HNE and HIO platforms, with the former measuring CFTR-channel activity as transepithelial currents and the latter measuring CFTR-mediated fluid influx into the organoid lumen, though a recent publication has demonstrated fluid flux measurements in HNE cell-derived organoids (Amatngalim et al, 2022).

3D HIO recapitulate the structural and functional motif of the in vivo tissue. Their use has been particularly valuable in CF research, where the combination of the simplicity of the forskolin-induced swelling (FIS) assay and the feature of an unlimited patient tissue resource made it a reliable preclinical predictive tool (Berkers et al, 2019; Ramalho et al, 2021; Aalbers et al, 2022). Transforming 3D HIO into epithelial monolayers (2D HIO) provided accessibility of the apical membrane which has so far been applied in studies examining host-pathogen and cell–nutrient interactions (Angus et al, 2019; Arnauts et al, 2022; Gunther et al, 2022). Recently, Zomer-van Ommen et al have successfully grown organoid-derived monolayers generated from pwCF and used them for functional CFTR assessment (Zomer-van Ommen et al, 2018). Furthermore, intestinal organoid-derived monolayers have been used for CFTR modulator drug testing in a single CF case study (Ciciriello et al, 2022). However, there has been no systematic evaluation or validation of 2D HIO for use in CFTR drug testing. Secondly, opening-up the intestinal organoids to monolayer allows transepithelial current measurements and thus direct comparison of the intestine and nasal cells as the two main tissues currently used for CFTR modulator drug testing.

In this study, we aimed to (a) directly compare electrophysiological characteristics in 2D HIO to HNE, (b) validate drug responsiveness of CFTR in 2D HIO against the current standard methods of HNE and 3D HIO, (c) evaluate clinical predictability of the in vitro CFTR drug responses measured in 2D HIO, and (d) assess feasibility of studying other apical membrane components relevant in CF.

# Results

## 2D HIO express higher CFTR gene expression and function compared to HNE

We first evaluated the epithelial structural and the baseline electrophysiological properties of the 2D HIO and compared these to the properties of the HNE cells. As shown in Fig 1A–C, 2D HIO cultures formed tight and polarized epithelial monolayers demonstrated by the presence of zonula occludens-1 (ZO-1) proteins and the expression of CFTR at the apical membrane. Transepithelial resistance measured with the Ussing chamber technique was 634 ± 202 $\Omega.cm^2$ (Rte; mean ± SD) for the non-CF control and 550 ± 342 $\Omega.cm^2$ for the CF cohort.

In comparison to HNE cells, HIO cells showed higher CFTR gene expression levels, consistent with previous reports of high CFTR gene expression in colonic epithelial cells (Fig 1D) (Strong et al, 1994). Therefore, it is not surprising that the magnitude of the CFTR-dependent ΔIeq (ΔFsk-Ieq and ΔCFTR$_{Inh-172}$-Ieq) responses were greater in 2D HIO compared with HNE cultures exemplified in 2D HIO generated from a non-CF individual and in 2D HIO from two *F508del/F508del* patients treated with tezacaftor/elexacaftor/ivacaftor (TEZ/ELEXA/IVA or VX-661+VX-445+VX-770) (Figs 1E and S1).

To study the responsiveness of the CFTR channel in 2D HIO to cAMP-dependent activation via the forskolin (Fsk) pathway, which in turn phosphorylates the regulator domain in the CFTR protein, we performed Fsk dose-response experiments. We observed increasing CFTR-mediated ΔIeq with increasing Fsk doses indicating similar phosphorylation-dependent CFTR channel activation in both HIO and HNE cells (Fig 1F–H). However, the range of the linear relationship between Fsk dose and CFTR activity was larger in 2D HIO compared with HNE cells reflecting, once again, the higher CFTR gene expression in the intestine.

## Measurable range of CFTR function is higher in 2D HIO compared to 3D HIO

To assess the dynamics of both Fsk-induced ion transport current and Fsk-induced fluid flux, we compared CFTR function measured as ΔFsk-Ieq in 2D HIO to the CFTR function measured as FIS in 3D HIO. We used intestinal organoids generated from a non-CF subject and TEZA/ELEXA/IVA CFTR-rescued intestinal organoids from two *F508del/F508del* patients (Fig 2A and B). In the TEZA/ELEXA/IVA-treated intestinal organoids from the *F508del/F508del* patients, the Fsk dose-response curves generated with the two different assays strikingly overlapped, capturing the parallel increase in organoid swelling via luminal fluid influx with incremental increase in CFTR-mediated ΔIeq (Fig 2B). In the non-CF individual, we observed an overall higher maximal CFTR response in the 2D HIO demonstrating linear Fsk activation of CFTR function until 10 μM Fsk in contrast to ~1 μM observed in 3D HIO (Fig 2A). This suggests that assessing the CFTR function in 2D HIO may help overcome the earlier reported limitation of the FIS assay in 3D HIO in measuring higher levels of the CFTR function (Dekkers et al, 2016). This observation further confirms that the limitation is because of the physics of the organoid swelling rather than the intestinal tissue culture itself.

difference between 2D HIO CFTR mRNA levels between non-CF versus *F508del/F508del CF* (Tukey's multiple comparison test, *P* = 0.11). **(E)** Transepithelial electrophysiological properties of 2D HIO and HNE measured in one non-CF subject (three technical replicates) and 2 *F508del/F508del* CF patients (two technical replicates each). Amil-amiloride (100 μM for HIO; 30 μM for HNE), Fsk-Forskolin (10 μM), CFTRinh-CFTR$_{Inh-172}$ (10 μM). Statistical analysis by one way ANOVA: Tukey's multiple comparison test, *P < 0.05, **P < 0.005, ****P < 0.0001. **(F)** Original traces show transepithelial current traces of the Fsk dose–response experiments assessing CFTR responses to increasing Fsk concentration (0.02–10 *μM*) and CFTRinh in 2D HIO and HNE from one non-CF subject (three technical replicates) and one *F508del/F508del* patient after chronic treatment with VX-661 (3 μM) + VX-445 (3 μM) and acute treatment with VX-770 (1 μM), experiments were done in two technical replicates. **(G, H)** Graphs compare the Fsk dose–response curves generated in non-CF and VX-661+ VX-445+ VX-770-treated *F508del/F508del* cultures between 2D HIO and HNE cells. Statistical test: Non-linear regression fit of the two dose–response curve and use of extra sum-of-squares F test to determine the statistical difference between the two curves, ***P < 0.0001.

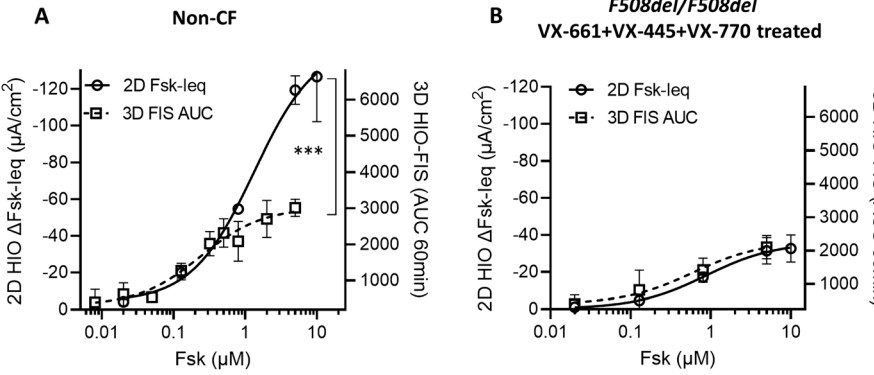

**Figure 2. Comparison of the forskolin-induced transepithelial current (2D HIO Ieq) and the forskolin-induced swelling (3D FIS) assessment in human intestinal organoids.**
**(A, B)** Graphs display Fsk dose–response curves assessing CFTR responses to increasing Fsk concentration (0.02–10 μM) performed in 2D HIO and 3D HIO in (A): a non-CF subject (three technical replicates for 2D HIO and four technical replicates for 3D HIO); (B): 2 *F508del/F508del* patients after chronic treatment with VX-661 (3 μM) + VX-445 (3 μM) +acute treatment with VX-770 (1 μM for 2D HIO, 3 μM for 3D HIO), (two technical replicates for 2D HIO and four technical replicates for 3D HIO for each patient). Statistical test: Non-linear regression fit of the two dose–response curve and the use of extra sum-of-squares F test to determine the statistical difference between the two curves, ***$P$ < 0.0001.

## 2D HIO discriminate responses to various CFTR modulators in pwCF with class I, II, and III mutations comparable to matched HNE and 3D HIO

We are the first to report a comparison between the modulator-induced CFTR responses measured in 2D HIO and those measured in HNE and 3D HIO within the same individual pwCF (Figs 3A and B and S2). HIO and HNE cells from pwCF homozygous for class I mutations were treated with a triple combination of TEZA/ELEXA/IVA (VX-661+VX-445+VX-770) with the addition of G418, a read-through agent, and SMG1i to inhibit nonsense-mediated decay. Our earlier work in HNE cells (Passage 2 [P2]) demonstrated that this drug combination can achieve some level of CFTR functional rescue (Laselva et al, 2020). In this study, we observed minimal responses in the 2D HIO, 3D HIO, and (P3) HNE cultures (Fig 3A and B).

For patients carrying class II CFTR mutations, preclinical and clinical studies have shown increased efficacy of the triple CFTR modulator drug TEZA/ELEXA/IVA, compared with the double combination of one corrector and one potentiator CFTR modulator drug (LUMA/IVA [VX-809 + VX-770]) (Keating et al, 2018; Veit et al, 2020). Similarly, we demonstrate that treatment of cultures from patients with at least one class II mutation with triple drug combinations led to larger increases in the CFTR function in 2D HIO when compared with double drug combinations. This increase in CFTR function was observed after treatment with either one of the following triple combination drugs; TEZA/ELEXA/IVA, AC-1/AC-2/AP2, PTI-428/PTI-801/PTI-808. 2D HIO demonstrated a similar response pattern to the different drugs and between different patients when compared with HNE and 3D HIO (Fig 3).

CFTR-potentiator-based treatment with IVA is approved for CF patients carrying at least one class III or IV mutation (Ramsey et al, 2011; Davies et al, 2013). Treatment with one of the potentiators, IVA or AP2, led to a significant increase in CFTR function with somewhat larger responses seen in 2D and 3D HIO, likely because of the higher CFTR expression, compared with HNE cultures (Fig 3). This difference could also be explained with tissue-specific potentiator effects and differences in CFTR assessment with HNE assessing immediate effects and 3D HIO cumulative effects over 60 min.

Because of the very fast uptake of clinical treatment with IVA among our pwCF with class III CFTR mutants, intestinal tissue samples were collected up to 6 mo post initiation of IVA treatment. In contrast, nasal epithelial cell samples that were collected as part of another research protocol and before commencing the treatment. Although we appreciate residual, likely drug-effect-related CFTR function in the native intestinal tissue specimen of these patients (Fig S3), residual CFTR function measured in intestinal organoids and in HNE cultures was relatively similar (Fig 3B), thus suggesting a drug wash-out during the process of intestinal organoid generation (see the Discussion section).

## Assessment in 2D HIO shows rescue of the N1303K CFTR gene variant by combinations of potentiators or triple drug combinations

There is a great interest in "theratyping" the *N1303K*-CFTR mutation, which is the fifth most common CFTR gene variant in the United States (US Patient Registry Annual report 2020, 2.4% of all CFTR gene variants). Its exact molecular rescue pathway still needs to be unraveled. Our previous work, confirmed by others, showed TEZA/ELEXA/IVA increased the N1303K CFTR function in HNE without increasing the abundancy of the CFTR protein at the membrane (Veit et al, 2021a; Laselva et al, 2021). This led to the understanding that ELEXA exerts both correcting and potentiating effects on mutant CFTR. Direct access to the apical membrane in *N1303K/N1303K* 2D HIO allowed us to study acute potentiators and multiple drug combinations offering better resolution of the sequential drug effects. As shown in Fig 4A and B, acute (at the time of Fsk addition) or chronic (18–24 h before experiments) application of IVA (VX-770) together with ELEXA (VX-445) significantly increased the *N1303K*-CFTR function in 2D HIO. Treatment with these two drugs achieved the same level of CFTR rescue compared with treatment with triple combination TEZA/ELEXA/IVA. Interestingly, sequential addition of IVA and ELEXA to the apical side resulted in a synergistic rescue effect on Fsk-activated *N1303K*-CFTR, independent of their order of application (Fig 4B).

## CFTR modulators effects in 2D HIO with class I, II and III CFTR mutations compared to matched HNE and 3D HIO

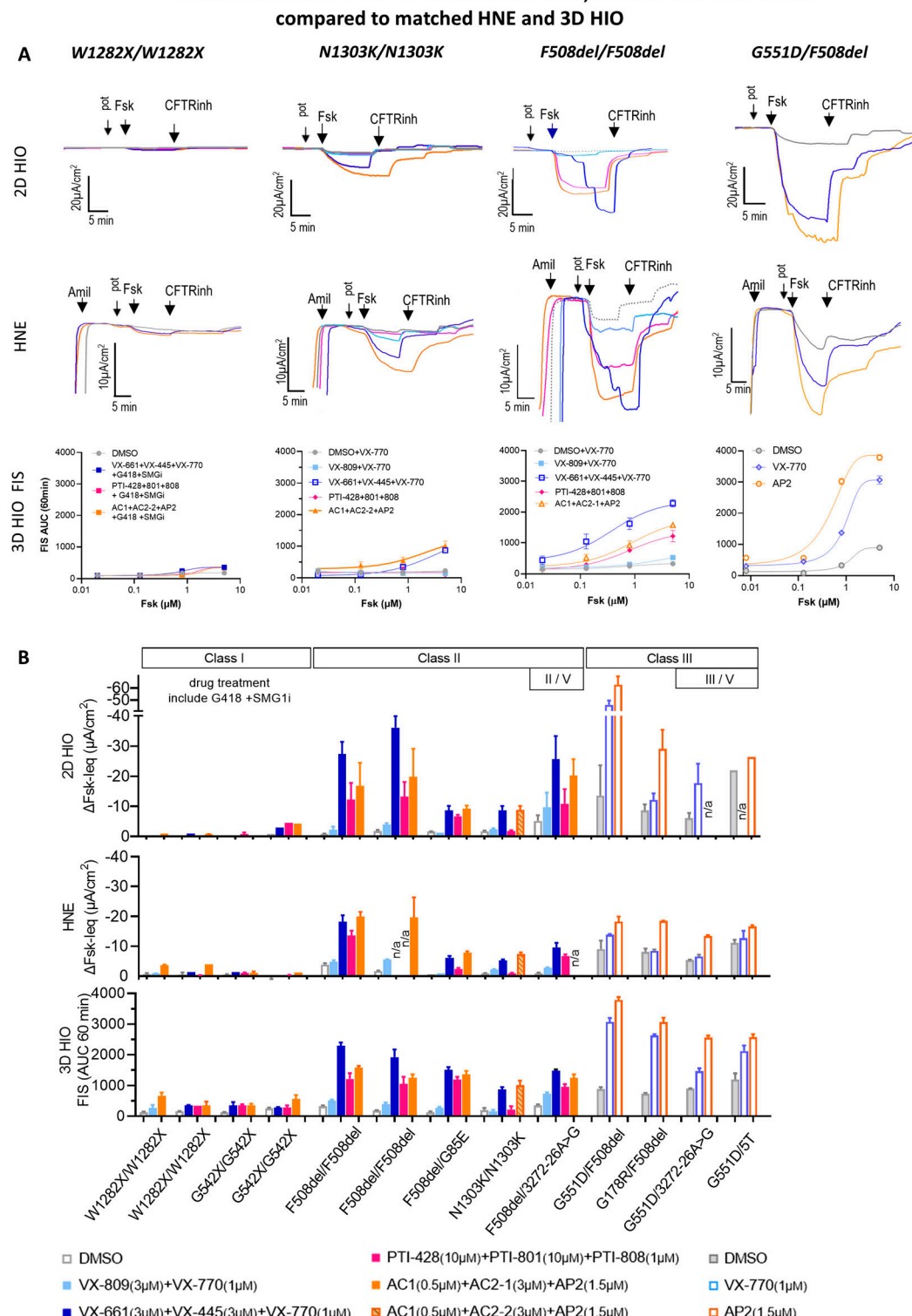

**Figure 3. CFTR modulators effects in 2D HIO with class I, II, and III CFTR mutations compared with matched HNE and 3D HIO.**
**(A)** Graphs show representative original transepithelial current (Ieq) traces from Ussing chamber studies on 2D HIO and HNE and the change of the area under the curve (FIS AUC) over 60 min in 3D HIO derived from CF patients with *W1282X/W1282X*, *N1303K/N1303K*, *F508del/F508del*, and *G551D/F508del* in response to various CFTR modulator drug treatments. Cultures were incubated with corrector drugs 18–24 h before Ussing experiments, and the potentiator drugs (pot) were applied before forskolin (Fsk) addition. Fsk - 10 μM/basolateral for 2D HIO and HNE, 5 μM for 3D HIO FIS; CFTRinh-172 - 10 μM/apical; amiloride (amil) - 100 μM/apical for 2D HIO, 30 μM/ apical for HNE. Potentiators were VX-770, PTI-808, and AP2. **(B)** Bar graphs summarize assessed CFTR function after the different CFTR modulator treatments comparing the

### In vitro response to CFTR modulator drugs measured in 2D HIO correlates with clinical response to therapy

Drug response data generated in HNE and 3D HIO showed good correlations between the in vitro response to CFTR modulator drugs and the clinical response (Pranke et al, 2017; Berkers et al, 2019; Ramalho et al, 2021; Aalbers et al, 2022). To see whether this is also the case using 2D intestinal monolayers, we evaluated for this correlation in a subset of pwCF for whom 2D HIO and clinical data on CFTR modulator treatment were available. Four patients with at least one class III CFTR allele were treated with IVA and the two patients with *F508del/F508del* received LUMA/IVA. In clinical trials, changes in SwCl and in forced expiratory volume in 1 s expressed as percent predicted value (FEV1%pred) are used as clinical measures of drug efficacy (Ramsey et al, 2011; Wainwright et al, 2015; Donaldson et al, 2018; Keating et al, 2018). The change in CFTR function measured in both 2D HIO (Fig 5A and B) and 3D HIO (Fig 5C and D) after treatment with IVA or LUMA/IVA, significantly correlated to the change in SwCl and FEV1%pred after 6 mo of CFTR modulator treatment. For one F508del/F508del, patient post-treatment sweat test was not available.

### Utilizing 2D HIO to study the epithelial sodium channel ENaC

Electrophysiological measurement of ENaC activity is quantified as the current response to inhibition with apically applied amiloride (ΔIeqAmil). We observed that the amiloride-sensitive ENaC current was near absent in our 2D HIO when compared with HNE (Figs 1E and S2). Other than the difference of submerged culture condition for our 2D HIO versus air–liquid interface (ALI) culture for HNE, another possible explanation of the minimal ENaC response is the absence of hydrocortisone in the intestinal cultural media. To test this theory, we compared the amiloride-Ieq response with and without hydrocortisone in the media. The addition of hydrocortisone to the media (4–5 d) resulted in the emergence of a significant amiloride-sensitive Ieq in non-CF and in CF 2D HIO cultures (Fig 6A and B).

## Discussion

In this study, we present comparative data to validate 2D HIO as a new preclinical tool for CFTR modulator drug testing and highlight the performance comparison between intestinal and nasal cell culture techniques. Because nasal and rectal tissues were sampled from a unique, but representative group of pwCF carrying Class I, II, III CFTR-mutations, this study will provide the necessary basis for future comparison of preclinical drug efficacy data among CF centres, as most of the centres have access to either nasal or intestinal tissue-based drug-testing platforms, but not to both. We found that 2D HIO favorably compared with matched electrophysiological Ussing chamber studies done in HNE cultures and to

FIS measurements performed in 3D HIO. Higher CFTR expression in the intestinal tissue and large range functional measurements without a ceiling effect as seen with 3D structures (Dekkers et al, 2013; Dekkers et al, 2016) are significant advantages of 2D HIO with respect to CFTR drug testing.

2D HIO have been used for various research purposes (Angus et al, 2019; Arnauts et al, 2022; Gunther et al, 2022) and as demonstrated in this study, the 2D HIO model is particularly relevant for evaluating the drug response in CF patients with various CFTR genotypes. Transforming the intestinal organoids into polarized intestinal epithelial monolayers allowed CFTR function assessment as transepithelial current measurements at baseline, and in response to various CFTR modulator treatments. Our comparative study showed an overall similar ranking pattern of the efficacy of CFTR modulator drugs and drug combinations in 2D HIO when compared with HNE and 3D HIO. Furthermore, it similarly delineated differences in CFTR drug responses between individual CF patients. For pwCF carrying nonsense CFTR gene variants on both alleles (class I), we did not observe changes in CFTR function despite adding a read-through agent (G418) and an inhibitor of nonsense-mediated decay (SMG1i) to the triple combination therapy across the techniques and tissues. Though this combination has shown some moderate rescue responses in passage 2 *W1282X* HNE cells previously (Laselva et al, 2020), newer readthrough agents are more promising in their ability to restore CFTR function (Sharma et al, 2021; de Poel et al, 2022) and will be studied in our system in the future. In tissue cultures derived from patients carrying at least one F508del-CFTR mutation, triple therapy (ELEXA/TEZA/IVA, PTI-428/801/808, or AC1/AC2-1/AP2) resulted in a larger CFTR-mediated response compared with double therapy displaying the same drug ranking pattern across all methods. These results are consistent with the results observed in clinical studies (Wainwright et al, 2015; Heijerman et al, 2019). 2D HIO were also very sensitive in detecting different CFTR responses to acute treatment with potentiators between different patients. Large potentiator effects in 2D HIO and 3D HIO, even though the intestinal tissue was retrieved while the patients were on IVA drug therapy, suggest a drug wash-out effect during the process of organoid generation and consequently allows the inclusion of pwCF on CFTR modulator therapy for future preclinical drug testing. Potentiator effect measured in the 2D and 3D HIO models was larger when compared with the HNE. This can be explained with the overall higher expression of CFTR in the intestinal tissue. Furthermore, differences in the binding mechanism of the potentiator to the CFTR protein at the membrane or differences in CFTR thermostability or gating capacity between nasal and intestinal tissues may be additional explanations for the observed differences. Evaluating tissue-specific binding sites and pharmacokinetics of the CFTR modulators may provide further insights into the observed differences in potentiator response and may help to explain differences in clinical organ response.

assessment in 2D HIO, HNE, and 3D HIO. Each bar represents one single patient with two technical replicates for class I and III patients and four technical replicates for class II patients. For tissue cultures from patients with class 1 mutations, G418 (read-through agent, 200 µg/ml) and SMGi (inhibitor of nonsense-mediated decay, 0.5 µM) were added to the corrector drugs 18–24 h before assessment. Note: HNE cell cultures of one patient with *F508del/F508del* failed the re-culturing process repeatedly after a freeze–thaw cycle leading to missing experimental data.

## N1303K CFTR rescue with the combination of VX-770+VX-445 or triple CFTR modulator combinations

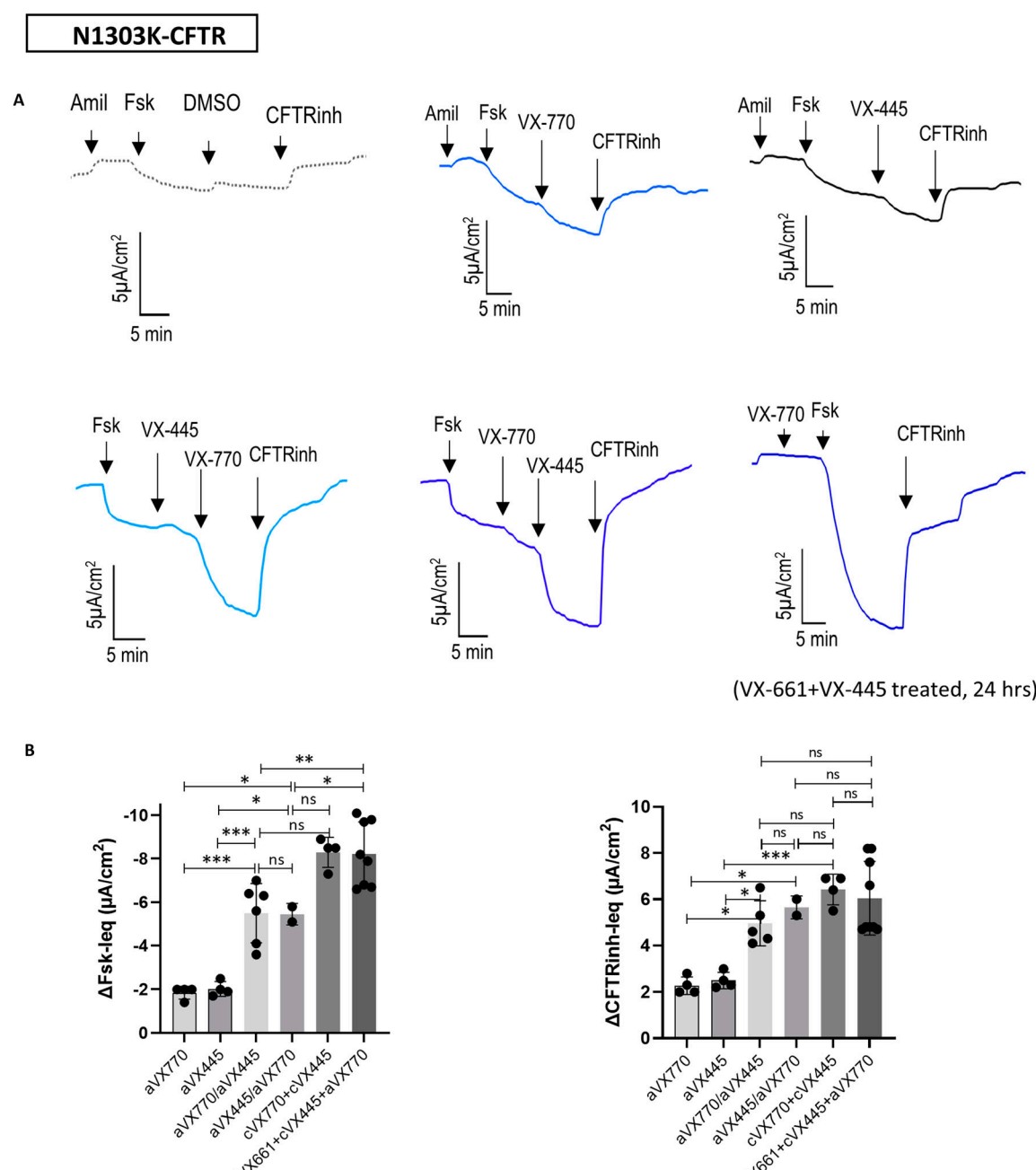

**Figure 4. N1303K CFTR rescue with the combination of VX-770+VX-445 or triple CFTR modulator combinations.**
**(A)** Original Ieq traces in 2D HIO from a *N1303K/N1303K* patient. CFTR modulator drugs were applied acutely at the time of Fsk (10 μM) stimulation to the apical side: VX-770 (1 μM), VX-445 (3 μM). CFTR$_{Inh-172}$-Ieq (CFTRinh, 10 μM) was added to confirm CFTR activity; chronic treatment with VX-661 (3 μM) and VX-445 (3 μM) was done over 24 h. Experiments were done in the presence of 100 μM amiloride. **(B)** Graphs summarize the drug-induced change in Fsk-Ieq and CFTRinh-Ieq after (1) acute addition of CFTR modulators apically after Fsk-induced CFTR activation: acuteVX-770 (aVX-770), acute VX-445 (aVX-445), (2) combination of aVX-770 and aVX-445 in different sequence of addition, and treatment with chronic incubation (18–24 h) of VX-770 (cVX-770)+ VX-445 (cVX-445), (3) triple drug combination chronic incubation of VX-661 and VX-445 with acute addition of VX-770 (cVX-661+cVX-445 +aVX-770). Results were obtained from two patients homozygous for the N1303K variant. Data are means ± SD, statistical analysis: Analysis of variance with Tukey's multiple comparison testing, *P < 0.05, **P < 0.005, ***P < 0.0005.

## In vitro CFTR functional response to CFTR modulatorS in 2D HIO correlated with in vivo clinical response

### 2D HIO versus in vivo response

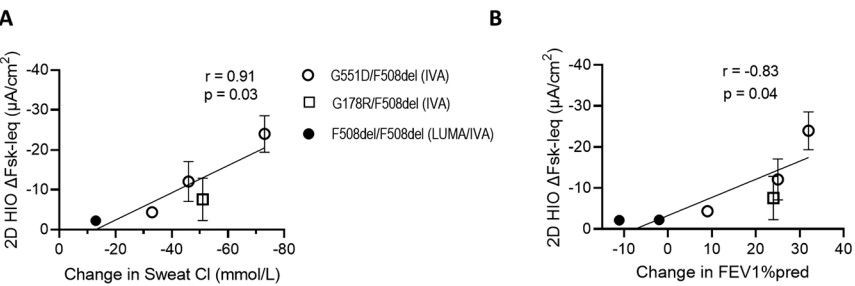

### 3D HIO versus in vivo response

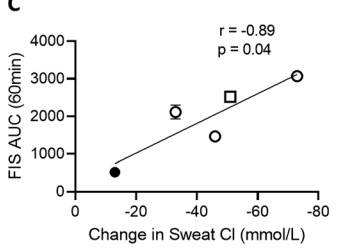
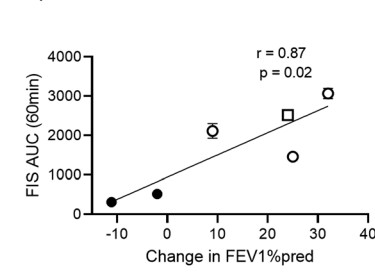

**Figure 5. In vitro CFTR functional response to CFTR modulators in 2D HIO correlated with in vivo clinical response.**
**(A, B, C, D)** Graphs show the correlation between the in vitro measured change in CFTR function after treatment with CFTR modulator drugs (VX-770 or IVA and VX-880 + VX-770 or LUMA/IVA) in 2D HIO (A, B) and in 3D HIO (C, D) on one side and clinical drug efficacy assessed as change in sweat chloride (Sweat Cl). **(A, B, C, D)** and change in FEV1%pred on the other (B, D). Fsk concentration for the 2D HIO experiments was 10 µM and 5 µM for the 3D FIS. Changes in the clinical markers are expressed as the difference between an assessment at baseline and 6 mo post treatment start. Each dot represents one pwCF. Open circles represent pwCF carrying at least one G551D allele (*G551D/F508del, G551D/3272-26A>G, G551D/5T*), open square represents one patient with *G178R/F508del* who started treatment with ivacaftor; closed circles represent pwCF carrying *F508del/F508del* alleles. This cohort contained 6 pwCF, however, post-treatment sweat chloride measurements were only available for 5 pwCF.

## Utilizing 2D HIO to study the apical membrane ENaC channel

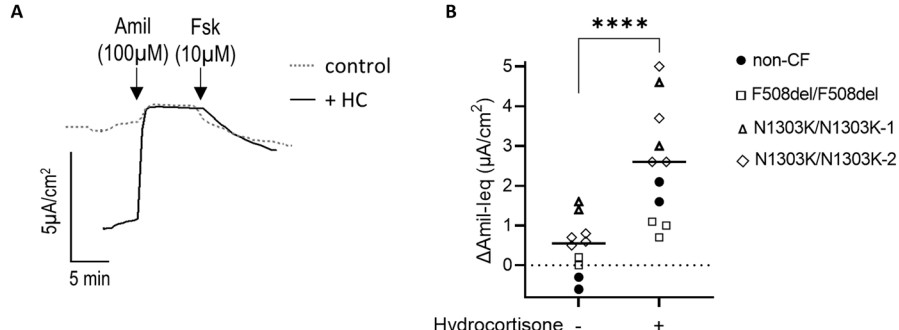

**Figure 6. Utilizing 2D HIO to study the apical membrane ENaC channel.**
**(A)** Representative original Ussing chamber Ieq traces using 2D HIO from an *N1303K/N1303K* patient to show the effect of pretreatment of 2D HIO with hydrocortisone (HC, 3 µg/ml) on amiloride-sensitive currents. The dotted grey line shows 2D HIO grown in culture media without hydrocortisone supplementation and the solid black line shows 2D HIO grown in culture media with hydrocortisone supplementation. **(B)** Graph summarizes paired experiments evaluating the Amil-sensitive Ieq in 2D HIO generated in growth media with and without hydrocortisone supplementation for 4–5 d. Statistical analysis was performed by *t* test, \*\**P* < 0.005.

The good correlation of CFTR function assessment between 2D HIO, HNE, and 3D HIO, similar CFTR channel activation kinetics, and good predictability of clinical outcome markers, demonstrate the viability of utilizing 2D HIO as a new drug testing platform.

A preclinical drug testing platform based on 2D HIO may even be advantageous. Rectal biopsies can be safely performed at all ages including infants (Ambartsumyan et al, 2020; Graeber et al, 2021) and intestinal organoids provide an unlimited tissue resource, making this method ideal for repeated and life-long preclinical drug testing. Good correlation between CFTR current measurements in 2D HIO and organoid swelling measurements in 3D HIO confirmed that the swelling of intestinal organoids in response to Fsk is driven by CFTR-mediated chloride currents, which was further illustrated by matching Fsk-dose–response curves in 2D and 3D HIO.

In contrast to 3D HIO which can underestimate large CFTR currents because of a mechanical limitation of the fluid influx (Dekkers et al, 2013, 2016), no such limitation was observed in 2D HIO. The opportunity of sequential application of Fsk as a CFTR-stimulating cAMP-agonist, and CFTR$_{Inh-172}$ as a CFTR inhibitor in 2D HIO provides an additional advantage and allows a more elaborate evaluation of CFTR drug responses. Lastly, access to the apical membrane and transepithelial measurements also allow discriminating between chloride and bicarbonate components of the CFTR function as recently reported by Ciciriello et al based on 2D intestinal monolayers from two pwCF with rare CFTR gene variants (Ciciriello et al, 2022).

Other apical membrane constituents play a role in CF pathophysiology and are of interest as alternative or adjunctive drug

**Table 1. CFTR modifying drugs.**

| Compound | Dose | Duration of culture treatment | Mechanism of action | Manufacturer |
|---|---|---|---|---|
| VX-770 ivacaftor (IVA) | 1 µM 3 µM (FIS) | acutely | CFTR potentiator (Van Goor et al, 2009; Phuan et al, 2019) | Selleckchem |
| VX-809 lumacaftor (LUMA) | 3 µM | 24 h | CFTR corrector (Van Goor et al, 2011; Okiyoneda et al, 2013) | Selleckchem |
| VX-661 tezacaftor (TEZA) | 3 µM | 24 h | CFTR corrector (Veit et al, 2020) | Selleckchem |
| VX-445 elexacaftor (ELEXA) | 3 µM | 24 h | CFTR corrector (Veit et al, 2020, 2021a) | MedChemExpress |
| AP2 X300529 | 1.5 µM, 2 µM (FIS) | acutely | CFTR potentiator (Laselva et al, 2020) | Abbvie Inc. |
| AC1 X281602 | 0.5 µM | 24 h | CFTR corrector (Kym et al, 2018; Laselva et al, 2020) | Abbvie Inc. |
| AC2-1 X281632 | 3 µM | 24 h | CFTR corrector (Kym et al, 2018; Laselva et al, 2020) | Abbvie Inc., North Chicago, IL, US |
| AC2-2 X300549 | 3 µM | 24 h | CFTR corrector (Kym et al, 2018; Laselva et al, 2020) | Abbvie Inc. |
| PTI-428 nesolicaftor | 10 µM | 24 h | CFTR amplifier (Giuliano et al, 2018; Dukovski et al, 2020) | Proteostasis Therapeutics |
| PTI-801 posenacaftor | 10 µM | 24 h | CFTR corrector (Kym et al, 2018) | Proteostasis Therapeutics |
| PTI-808 dirocaftor | 1 µM | 24 h | CFTR potentiator (Kym et al, 2018) | Proteostasis Therapeutics |
| G418 | 200 µg/ml | 24 h | Read-through agent (Laselva et al, 2020) | Sigma-Aldrich; G8168 |
| SMG1i | 0.5 µM | 24 h | Non-sense mediated decay inhibitor (Laselva et al, 2020) | CF Foundation Therapeutics Lab |

targets (Moore & Tarran, 2018; Kota, 2022; Lim et al, 2022), and these can be assessed using 2D HIO. CF researchers have been particularly interested in targeting the epithelial sodium channel (ENaC) as it is thought of as a major co-player in CF lung disease. We attributed the initial absence of measurable ENaC channel activity in 2D HIO to a lack of hydrocortisone in the intestinal culture media, because rectal tissue expresses ENaC (Kunzelmann & Mall, 2002) and glucocorticoid and mineralocorticoid receptors are present in the colon influencing ENaC expression (Kunzelmann & Mall, 2002). In fact, addition of hydrocortisone to the intestinal culture media significantly increased ENaC activity in 2D HIO to measurable levels making this tissue amenable for ENaC-focused studies as well (Moore & Tarran, 2018; Kota, 2022). Next to ENaC, several apically localized solute carrier transporters and other membrane bound proteins have been identified to closely interact with CFTR and are currently under further investigations as potential drug targets in CF (Sun et al, 2012; Ahmadi et al, 2018; Lim et al, 2022). 2D HIO offer the opportunity to study the functional interaction between CFTR–network proteins more accurately, as recently shown by Lim et al who reported an increase in F508del-CFTR-mediated currents upon knock-down of the apical membrane bound fibrinogen-like 2 (FGL-2) protein (Lim et al, 2022).

Finally, we showcased the utility of 2D HIO monolayers in obtaining further insight into the mechanism of CFTR drug modulation using the example of the *N1303K*-CFTR gene variant. The *N1303K* gene variant is categorized as a class II mutation with a molecular defect in the second nucleotide-binding domain (NBD2) of the CFTR protein but does not exhibit the expected protein rescue effect upon triple combination treatment (Laselva et al, 2021). Using 2D HIO from 2 pwCF homozygous for *N1303K*, we

observed a rescue response to treatment with ELEXA/TEZA/IVA as shown previously in HNE cultures (Laselva et al, 2021). In addition, and in agreement with previous studies, we showed a moderate rescue effect with the combined treatment of dual CFTR modulator treatment with ELEXA and IVA reaching a similar level of CFTR functional response compared with triple combination therapy (Phuan et al, 2019; Son et al, 2021; Ensinck et al, 2022). Using a reversed sequence of modulator application (IVA and ELEXA) in 2D HIO, this synergistic effect is shown to be interdependent between ELEXA and IVA and is not mainly driven by IVA as previously believed (Laselva et al, 2021). IVA elicits its potentiating effect by increasing the open probability of the CFTR channel (Langron et al, 2018), but the mechanism of ELEXA is not yet understood. Nevertheless, our data suggest that both drugs interact at different sites of the CFTR protein to synergistically enhance CFTR channel conductance, which has also been suggested by Veit et al (2021a). 2D HIO are an excellent tool to further understand patient-specific molecular defects and to optimize CFTR-targeted treatment.

### Limitation of this study

The main limitation of this study is the sample size. Nevertheless, we present a unique group of pwCF including rare CFTR gene variants and for the first time report responses to various CFTR modulator drug combinations. Furthermore, only a small number of patients were started on CFTR modulator treatment during the period of this study limiting our in vitro–in vivo analysis. This is mainly because of the delay in Health Canada approval of the triple drug combination and restricted funding available for the dual drug combination in

Canada. Despite this limitation, we were able to show good correlation between clinical response and 2D HIO in vitro drug testing.

# Materials and Methods

### Study cohort

This study was approved by the Research Ethics Review Board of the Hospital for Sick Children, Toronto, Canada (REB # 1000058992). After informed consent, rectal biopsies were obtained using biopsy forceps. Nasal cell data were obtained from the Canada-Sick Kids Program for Individualized CF Therapy (CFIT): https://lab.research.sickkids.ca/cfit/ (REB# 1000044783), and may have been reported before for some patients (Laselva et al, 2020, 2021). A total of 15 individuals were recruited, one non-CF individual and 14 pwCF with the following mutations: *W1282X/W1282X* (two patients), *G542X/G542X* (two patients), *F508del/F508del* (two patients), *N1303K/N1303K* (two patients) *F508del/G85E, F508del/3272-26A→G, F508del/G551D, G551D/3272-26A→G, G178R/F508del,* and *G551D/5T*. For some pwCF, clinical data were available 6 mo after starting therapy which were collected as part of an observational study (REB #1000036334). Demographics and clinical characteristics of the participants are shown in Table S1.

### Intestinal organoids—2D monolayer

The 2D HIO culture method was adopted from a previously published protocol using patient-derived intestinal organoids generated from fresh rectal biopsies (Zomer-van Ommen et al, 2018; Vonk et al, 2020). Mature 3D HIO were dissociated with trypsin (TrypLE; Thermo Fisher Scientific), diluted with growth medium (IntestiCult Human OGM, STEMCELL Technologies) supplemented with Y-27632, and seeded onto Purecol pre-coated transwell inserts (Costar 3470, 6.5 mm diameter, 0.4 $\mu$m pore size; Corning) for generating a polarized monolayer (Zomer-van Ommen et al, 2018). Y-27632 was removed 1 d post-seeding and the growth media were changed every other day. Ussing chamber studies were conducted 4–5 d post-seeding. Cultures were incubated with CFTR modulators (Table 1) 18–24 h before Ussing experiments.

### HNE cell cultures

HNE cultures were established as previously described (Eckford et al, 2019). Briefly, after nasal brushing, nasal epithelial cells were grown in PneumaCult EX medium (STEMCELL Technologies) supplemented with antibiotics and expanded. Cells from passage (P) 3 were ultimately used for experiments after 14 d of differentiation in ALI with PneumaCult ALI medium (STEMCELL Technologies).

### Forskolin-induced swelling (FIS) assay

Forskolin (Fsk)-induced swelling assays were performed as previously described (Vonk et al, 2020). Organoids were stimulated acutely with Fsk titration doses of 0.02, 0128, 0.8, 5.0 µM (for class I

mutations: 0.8, 5.0 µM Fsk). 3D HIO were incubated for 18–24 h with CFTR modulators before the FIS assays. Images were captured using live-cell imaging microscopy (Cellomics ArrayScan VTI HCS Reader at 2.5X magnification; Thermo Fisher Scientific) at 37°C for 60 min at 10-min intervals and quantified using a custom-made MATLAB-based software program (GUI v2.15, 2019). The average area under the curve (AUC) of the mean change (±SE, n = 2–3 technical replicates) of the total organoid surface area at t = 60 min after the addition of Fsk was used as the CFTR functional read-out (Vonk et al, 2020).

### Ussing chamber experiments

Electrophysiological measurements using circulating Ussing chamber system (EM- CSYS-4; Physiologic Instruments), in open circuit mode were performed as previously described (Laselva et al, 2021) with symmetrical chloride–bicarbonate buffer. CFTR function was quantified as the change in transepithelial current upon addition of Fsk ($\Delta$Fsk-Ieq, 10 µM/basolateral) and confirmed by the magnitude of CFTR inhibition with CFTRinh-172 ($\Delta$CFTRinh-Ieq 10 µM/apical, second dose applied consequently when large Fsk responses were observed). Experiments were performed in the presence of amiloride (100 µM for 2D HIO, 30 µM for HNE). Ussing data acquisition and analysis were completed using a custom-made LabVIEW-based program (UCP4.4.1, 2015).

### Immunocytochemistry

2D HIO were fixed in 4% paraformaldehyde and incubated with the following primary antibodies: mouse anti-CFTR, clone 13-1 (R & D Systems), 1:200 dilution; rabbit anti-zonula occludens-1 (ZO1; Invitrogen), 1:300 dilution in 5% of BSA-PBS; polyclonal goat anti-Ezrin (Santa Cruz), 1: 100 dilution, and then secondary antibodies: Alexa 488 or 594 HRP conjugated to anti-mouse IgG, anti-rabbit IgG (Invitrogen); diluted 1:400 in 5% BSA-PBS. DAPI (4′,6-diamidino-2-phenylindole; Invitrogen) was used to stain the nuclei. Samples were imaged with Olympus spinning disc confocal microscope (Du et al, 2015).

### Quantitative real-time PCR (qRT-PCR)

After extraction of RNA (Illustra RNA spin Mini RNA Isolation kit), a sufficient RNA concentration of >100 ng/µl was confirmed by measurements with the NanoDrop One (Thermo Fisher Scientific). Next, 1 µg of total RNA was treated with ezDNase enzyme (Thermo Fisher Scientific) to remove genomic DNA and used for cDNA synthesis (iSCRIPT cDNA synthesis kit, Bio-Rad). Quantitative real-time PCR was performed using SsoFast EvaGreen supermix (Bio-Rad), gene-specific primers and Biorad CFX Connect Real-Time PCR Detection System. Expression levels were determined with duplicate assays per sample and normalized to 18S ribosomal RNA (Lim et al, 2022).

### CFTR modulators

CFTR modulator drugs from Vertex Pharmaceuticals approved for clinical treatment were used for this study. Furthermore, we used

CFTR modulators from Galapagos/AbbVie and from Proteostasis Therapeutics, now governed under FAIR therapeutics, which have been tested preclinically and/or clinically in phase I, II, III clinical trials (https://www.cff.org/Trials/Pipeline) (Bell et al, 2020; Dukovski et al, 2020; de Poel et al, 2021). Table 1 lists the CFTR modulator drugs used in this study.

## Statistics

All data are presented as mean ± SD. $t$ test, one- or two-way ANOVA and Tukey`s multiple comparison test were used to evaluate differences between groups using a significance level of 0.05. Pearson's correlation was used for correlation analysis between assay techniques and in vitro–in vivo correlations. Non-linear regression was used to compare dose–response curves and the extra sum-of-squares F test to determine the statistical difference between them. All analyses were performed using GraphPad Prism 9.0.

# Data Availability

Data, analytic methods, and study materials can be made available to individual researchers upon request to the corresponding author.

# Supplementary Information

# Acknowledgements

We thank Abbvie Inc and Proteostasis/FAIR Therapeutics for sharing their CFTR modulator drugs for this study. We thank Cathleen Duan for assistance in RNA extraction from HIO and HNE cultures. Furthermore, a very special thanks goes out to all the participants of this study.

## Author Contributions

L Birimberg-Schwartz: conceptualization, resources, data curation, formal analysis, investigation, visualization, methodology, project administration, and writing—original draft, review, and editing.
W Ip: conceptualization, resources, data curation, formal analysis, investigation, visualization, methodology, and writing—original draft, review, and editing.
C Bartlett: resources, data curation, formal analysis, investigation, methodology, and writing—review and editing.
J Avolio: resources, data curation, project administration, and writing—review and editing.
AM Vonk: methodology and writing—review and editing.
T Gunawardena: resources, data curation, investigation, and writing—review and editing.
K Du: data curation, investigation, and writing—review and editing.

M Esmaeili: data curation, investigation, and writing—review and editing.
JM Beekman: methodology and writing—review and editing.
J Rommens: methodology and writing—review and editing.
L Strug: resources and writing—review and editing.
CE Bear: resources, methodology, and writing—review and editing.
TJ Moraes: resources, methodology, and writing—review and editing.
T Gonska: conceptualization, resources, formal analysis, supervision, funding acquisition, visualization, methodology, project administration, and writing—original draft, review, and editing.

## Conflict of Interest Statement

T Gunawardena performed consultation service for Vertex Pharmaceuticals. T Gonska received funding from Vertex Pharmaceuticals for an investigator-initiated study and from AbbVie as part of a collaborative research project.

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
