## [Reviewer comments · Life Science Alliance]

Life Science Alliance

Validating organoid-derived human intestinal monolayers for personalized therapy in cystic fibrosis

Liron Birimberg-Schwartz, Wan Ip, Claire Bartlett, Julie Avolio, Annelotte Vonk, Tarini Gunawardena, Kai Du, Mohsen Esmaeili, Jeffrey Beekman, Johanna Rommens, Lisa Strug, Christine Bear, Theo Moraes, and Tanja Gonska

DOI: <https://doi.org/10.26508/lsa.202201857>

Corresponding author(s): Tanja Gonska, Hospital for Sick Children and Liron Birimberg-Schwartz, The Hospital for Sick Children, University of Toronto

Review Timeline:

Submission Date:	2022-11-26
Editorial Decision:	2023-01-17
Revision Received:	2023-02-24
Editorial Decision:	2023-03-20
Revision Received:	2023-03-27
Accepted:	2023-03-28

Scientific Editor: Novella Guidi

Transaction Report:

January 17, 2023

Re: Life Science Alliance manuscript #LSA-2022-01857-T

Tanja Gonska
Hospital for Sick Children
555 University Avenue
Toronto, Ontario M5G 1X8
CANADA

Dear Dr. Gonska,

Thank you for submitting your manuscript entitled "Validating organoid-derived human intestinal monolayers for personalized therapy in cystic fibrosis" to Life Science Alliance. The manuscript was assessed by expert reviewers, whose comments are appended to this letter. We invite you to submit a revised manuscript addressing the Reviewer comments.

Thank you for this interesting contribution to Life Science Alliance. We are looking forward to receiving your revised manuscript.

Sincerely,

B. MANUSCRIPT ORGANIZATION AND FORMATTING:

Reviewer #1 (Comments to the Authors (Required)):

The aim of the study was to compare the electrophysiological evaluation of CFTR function in 2D Human Intestinal Organoids (HIO) to the validated models of Human Nasal Epithelial cells (HNE) and the 3D HIO. The authors wanted to validate the electrophysiological data obtained in response to various combinations of drugs approved for CFTR correction and potentiation. They mostly intended to demonstrate that the 2D HIO has less limitations than the 3D version by providing access to the apical side of intestinal cells, thus enabling more thorough evaluation of drugs on CFTR function and other membrane proteins. Finally, they authors wanted to demonstrate the validity of these methods, and in particular the 2D HIO, for the pre-clinical assessment of CFTR modulators by comparing in vitro data with clinical outcomes.

The authors acknowledge the study limitations, and in particular the very small number of samples available for each group of mutations.

In general, the conclusions of the study are well supported by the data presented. The methods used are appropriate to answer the questions and the comparative data are convincing.

The weakest part of the study is the correlation between the in vitro data (electrophysiological measurements of CFTR function) with the patient outcomes such as FEV1 or sweat chloride values. Although very interesting, the direct assessment of correlation seems like a big stretch. Are there any evidence that this correlation is valid or approved for clinical assessment of CFTR modulators?

A minor point is that the paper is written for the specialist and may be difficult to follow for anyone not familiar with the field.

Reviewer #2 (Comments to the Authors (Required)):

Ms LSA-2022-01857-T Life Science Alliance

Overview: In this manuscript, from an excellent team comprised of multiple individual labs, the authors set out to provide a comparison of the efficacy of multiple CFTR modulator treatments when assessed across three different platforms: (a) primary human nasal epithelial cells (HNEs) in traditional culture, (b) 3D human intestinal organoids (3D-HIOs), and (c) 3D-HIOs converted into 2D format (2D-HIOs), thus enabling study by electrophysiology in a manner identical to the HNEs. The concept of this study is a good one, and the comparisons generated may turn out to be useful. However, there are several concerns that must be addressed.

Major comments:

1. Page 7, bottom. This is pretentious. You may be the first to report this comparison, but you can't know if you are the first to make the comparison.
2. Figure 3, legend. What does "aVX-770" mean in this figure and the next? What does "cVX-661" mean? Figures and figure legends must be able to stand alone without reference to the full text. But these definitions don't exist in the full text, either.
3. Figures 1, 3, legend, and others. What are doses of Amil, Fsk, and 172?
4. Figure 3A, data for HNEs bearing F508del/F508del. Why does the VX-770 trace look so odd here? It really throws off the ability to make comparisons. Were multiple doses of VX applied, or multiple Fsk doses?
5. Figure 3B. Why are we missing the data for HNEs bearing F508del/F508del? These seem really crucial to supporting your general argument, and you likely already have those data from this exceedingly common genotype.
6. Figure 3B. The authors have paid scant attention to discussing cases where the results across the three platforms do NOT agree, such as for the Class III mutants comparing 3D-HIOs to HNEs.
7. Figure 4. I really, really don't like this analysis. Really don't like this analysis. Within a treatment group, essentially none of these are linear. You probably only get a decent fit by including all of the treatment groups on one graph. That doesn't seem like the right thing to do.
8. Figure 5, legend. This is very misleading. None of these experiments include "double potentiators".
9. Figure 6, legend. This last segment, RE PwCF homozygous for F508del, is really concerning. Are you telling us that you only included data for subjects for whom there was a significant correlation, and then including these data points in calculation of the

overall correlation? Why is the number of data points different in A and C vs. B and D.
10. Figure 7, legend. Since when is ENaC a "transporter"?

Responses to Reviewers**Reviewer #1 (Comments to the Authors (Required)):**

In general, the conclusions of the study are well supported by the data presented. The methods used are appropriate to answer the questions and the comparative data are convincing.

The weakest part of the study is the correlation between the in vitro data (electrophysiological measurements of CFTR function) with the patient outcomes such as FEV1 or sweat chloride values. Although very interesting, the direct assessment of correlation seems like a big stretch. Are there any evidence that this correlation is valid or approved for clinical assessment of CFTR modulators?

Response: Unlike the US, Canadian provinces did not fund the new CFTR modulator drug lumacaftor/ivacaftor which was the first drug available for CF patients with F508del/F508del. Thus, we only had a limited number of data to show some in vitro- clinical correlation. Sermet-Gaudelus group demonstrated correlation between CFTR functional change to CFTR modulator drugs measured in nasal cells and the change in clinical parameters of patients on this drug treatment (Sci Rep 2017, PMID 28785019). Similar in vitro-clinical response correlation was shown for intestinal organoids by a European group (Eur Resp J 2021, PMID

32747394). Thus, the concept that in vitro assessment using nasal and intestinal cells of individual patients predicts correlates with drug-induced clinical improvement already exists. We added our data to confirm that the same is true for 2D organoids and acknowledged the limitation of the small sample size.

A minor point is that the paper is written for the specialist and may be difficult to follow for anyone not familiar with the field.

Response: We have gone through the manuscript and edited it to make it more understandable for a more general audience.

Reviewer #2 (Comments to the Authors (Required)):

Ms LSA-2022-01857-T Life Science Alliance

Major comments:

1. Page 7, bottom. This is pretentious. You may be the first to report this comparison, but you can't know if you are the first to make the comparison.

Response: This is an important distinction. We have changed the wording accordingly as suggested.

2. Figure 3, legend. What does "aVX-770" mean in this figure and the next? What does "cVX-661" mean? Figures and figure legends must be able to stand alone without reference to the full text. But these definitions don't exist in the full text, either.

Response: Thank you for pointing this out. This was an oversight from our part. We have removed the "a" and "c" prefixes to avoid confusion, and added an explanation on the timing of the drug treatments.

3. Figures 1, 3, legend, and others. What are doses of Amil, Fsk, and 172?

Response: We have added the concentrations in all the relevant figures or figure legends.

4. Figure 3A, data for HNEs bearing F508del/F508del. Why does the VX-770 trace look so odd here? It really throws off the ability to make comparisons. Were multiple doses of VX applied, or multiple Fsk doses?

Response: We have improved the lining up of the traces. Now it is easier to observe that not the VX-770 traces, but rather the VX-661+VX-445+VX-770 trace in the 2D HIO and HNE F508del/F508del cultures are different. This is because these were used to generate our Fsk dose-response analysis.

5. Figure 3B. Why are we missing the data for HNEs bearing F508del/F508del? These seem

really crucial to supporting your general argument, and you likely already have those data from this exceedingly common genotype.

Response: Infrequently, nasal cell cultures fail in our hands. The nasal cells of this patient matching the rectal biopsies grew well until passage P2 and P3 three years ago. At that time point elexacaftor/tezacaftor/ivacaftor was not available for laboratory use, which explains the missing experiments. Following a freeze and thaw cycle our team was unable to grow these cells into P3 air-liquid interface (ALI) when this drug combination became available. We have made every effort to establish nasal cultures from the said individual by carrying out nasal brushes for a second time which did not lead to measurable cell cultures. In response to the reviewer's comment, we have tried one more time to thaw this patient's nasal cells for re-culturing into P3 ALI. We observed poor revival efficiency of the frozen cells of this individual, despite increasing the seeding density.

We have debated whether to discard this patient all together from our results due to the missing data, but found that the response to lumacaftor/ivacaftor as well as Abbvie combination which is available for all 3 methods still provides useful information.

We included an explanation about the absence of this experiment in the legend of Figure 3.

6. Figure 3B. The authors have paid scant attention to discussing cases where the results across the three platforms do NOT agree, such as for the Class III mutants comparing 3D-HIOs to HNEs.

Response: Thank you for your comment. We have expanded on these differences in the results and discussion as follows:

“CFTR-potentiator based treatment is approved for CF patients carrying at least one class III or IV mutation (Davies *et al.*, 2013; Ramsey *et al.*, 2011). Treatment with one of the potentiators, IVA or AP2, led to a significant increase in CFTR function with somewhat larger responses seen in 2D and 3D HIO, likely due to the higher CFTR expression, compared to HNE cultures (**Fig 3**). This difference could also be due to tissue-specific potentiator effects, as well as differences in CFTR assessment with HNE assessing immediate effects and 3D HIO cumulative effects over 60 minutes (page 9).” “Potentiator effect measured in the 2D and 3D HIO model were larger when compared to the HNE. This can be explained with the overall higher expression of CFTR in the intestinal tissue. Furthermore, differences in the binding mechanism of the potentiator to the CFTR protein at the membrane, or differences in CFTR thermostability or gating capacity between nasal and intestinal tissue, may be additional explanations for the observed differences. Evaluating tissue-specific binding sites and pharmacokinetics of the CFTR modulators may provide further insights into the observed differences in potentiator response and may help to explain differences in clinical organ response (page 13).”

Despite the differences in the magnitude of response across the methods, the ranking of drugs according to their rescue effect on the cultures remained similar. This emphasizes that these culture systems can be used interchangeably in CF personalized medicine.

7. Figure 4. I really, really don't like this analysis. Really don't like this analysis. Within a treatment group, essentially none of these are linear. You probably only get a decent fit by

including all of the treatment groups on one graph. That doesn't seem like the right thing to do.

Response: We did get a good fit even when analysing only one treatment group. These graphs were meant to underscore correlation between these assessments. While there may always be some difference between methodologies, in general all these 3 methods agree in terms of ranking best drug response. As discussed, magnitude differences are mainly due to a) higher CFTR expression in the intestine, b) capacity limit of the FIS assay. However, we accept the reviewers' point of view on this figure and have decided to remove it from the manuscript.

8. Figure 5, legend. This is very misleading. None of these experiments include "double potentiators".

Response: Thank you for pointing this out, we changed the wording to reduce the confusion. However, what was meant is that VX-445 and VX-770, both were applied acutely at the time of Fsk addition in the N1303K/N1303K cell cultures, suggesting co-potential of CFTR as previously reported (Laselva et al. Eur R J 2021 and Veit G et al. JCF 2021).

9. Figure 6, legend. This last segment, RE PwCF homozygous for F508del, is really concerning. Are you telling us that you only included data for subjects for whom there was a significant correlation, and then including these data points in calculation of the overall correlation? Why is the number of data points different in A and C vs. B and D.

Response: As responded to the first reviewer, Orkambi (lumacaftor and ivacaftor), was not funded by public funding agencies and thus out of reach for most of the Canadian patients with CF. For the analysis presented in these graphs, we used all the data available at that time.

All CF patients who had undergone intestinal organoid testing and who commenced CFTR modulator therapy were included in this analysis. There was no selection. Mismatch of numbers (6 for lung function correlation and 5 for sweat test correlation) is due to missing post-treatment sweat test results in one patient with F508del/F508del who came from a different hospital.

We added a sentence in this Results section and relevant figure legend to clarify.

“For one F508del/F508del patient post-treatment sweat test was not available (page 11).”

10. Figure 7, legend. Since when is ENaC a "transporter"?

Response: Thank you. We corrected this.

March 20, 2023

RE: Life Science Alliance Manuscript #LSA-2022-01857-TR

Tanja Gonska
Hospital for Sick Children
555 University Avenue
Toronto, Ontario M5G 1X8
CANADA

Dear Dr. Gonska,

Thank you for submitting your revised manuscript entitled "Validating organoid-derived human intestinal monolayers for personalized therapy in cystic fibrosis". We would be happy to publish your paper in Life Science Alliance pending final revisions necessary to meet our formatting guidelines.

- please address Reviewer 2's remaining comments
- please add ORCID ID for both corresponding authors-you should have received instructions on how to do so
- please add the Twitter handle of your host institute/organization as well as your own or/and one of the authors in our system
- the Summary after the Discussion section should be removed

A. FINAL FILES:

B. MANUSCRIPT ORGANIZATION AND FORMATTING:

Sincerely,

Reviewer #2 (Comments to the Authors (Required)):

The authors are commended for their excellent attention to the requested edits in my prior review. Quite responsive.

Two minor things need attention:

- 1) In Fig. 4 legend, need to know what p value corresponds to three asterisks.
- 2) In Fig. 6 legend, need to know the CFTR genotype for the cells used for both traces. If they are not the same, then this is not a fair comparison, given the clear genotype-dependence shown in the summary data at right.

-please address Reviewer 2's remaining comments

Response: These comments were addressed as follows.

Two minor things need attention:

1) In Fig. 4 legend, need to know what p value corresponds to three asterisks.

Response: The p value (<0.0005) was added to the legend.

2) In Fig. 6 legend, need to know the CFTR genotype for the cells used for both traces. If they are not the same, then this is not a fair comparison, given the clear genotype-dependence shown in the summary data at right.

Response: Both traces presented on the left (Figure 6A) are from experiments performed on the same N1303K/N1303K organoid cultures. We added clarifications in the figure legend accordingly.

-please add ORCID ID for both corresponding authors-you should have received instructions on how to do so

Response: ORCID ID was added

-please add the Twitter handle of your host institute/organization as well as your own or/and one of the authors in our system

Response:

These were added as requested.

-the Summary after the Discussion section should be removed

Response: This summary was removed.

In addition:

All necessary files have been uploaded to the system including editable version of the final text and figures.

The summary blurb is in the main text as advised.

The manuscript was prepared according to the instructions in the author's guidelines. Of course, we will be happy to make necessary changes if something was overlooked.

March 28, 2023

RE: Life Science Alliance Manuscript #LSA-2022-01857-TRR

Dr. Tanja Gonska
Hospital for Sick Children
555 University Avenue
Toronto, Ontario M5G 1X8
Canada

Dear Dr. Gonska,

Thank you for submitting your Research Article entitled "Validating organoid-derived human intestinal monolayers for personalized therapy in cystic fibrosis". It is a pleasure to let you know that your manuscript is now accepted for publication in Life Science Alliance. Congratulations on this interesting work.

DISTRIBUTION OF MATERIALS:

Again, congratulations on a very nice paper. I hope you found the review process to be constructive and are pleased with how the manuscript was handled editorially. We look forward to future exciting submissions from your lab.

Sincerely,
